# Multimodal prehabilitation in people awaiting acute inpatient cardiac surgery: Study protocol for a pilot feasibility trial (PreP-ACe)

Sarah Raut[1]*, Aaron Hales[1], Maureen Twiddy[2], Lili Dixon[1], Dumbor Ngaage[1], David Yates[3,4], Gerard Danjoux[4,5], Lee Ingle[6]

**1** Hull University Teaching Hospitals NHS Trust, Hull, United Kingdom, **2** Hull and York Medical School, University of Hull, Hull, United Kingdom, **3** York and Scarborough Teaching Hospitals NHS Foundation Trust, York, United Kingdom, **4** North Yorkshire Academic Alliance of Perioperative Medicine, North Yorkshire, United Kingdom, **5** South Tees Hospital N.H.S. Foundation Trust, Middlesbrough, England, **6** Faculty of Health Sciences, School of Sport, Exercise & Rehabilitation Science, University of Hull, Hull, United Kingdom

* sarah.raut@nhs.net

## Abstract

### Background

The concept of "prehabilitation," or optimising individual physical and mental wellbeing prior to surgery is well established in cancer and orthopaedic populations. However, amongst the cardiac surgery population, the concept is relatively new. Of the few studies available, all focus on the elective surgical population. This pilot feasibility trial is novel as it will focus on the impact of multimodal prehabilitation on the acute inpatient cardiac surgical population.

### Methods

This single centre, prospective, single arm pilot feasibility trial will recruit 20 inpatients awaiting cardiac surgery. Measurements will be collected at the start of the trial (baseline), 7 days after intervention, and 14 days after the intervention or before the day of surgery. The primary outcome measure will be feasibility and practicality of the programme in an acute inpatient population. We will be looking into participant eligibility, acceptability, recruitment rates, completion rates and barriers to implementing a prehabilitation programme. Secondary outcomes include incidence of study-related adverse events, improvement in 6 minutes walk test (6MWT), hand grip strength, quality of life, anxiety scores and spirometry. At the end of the trial, we will be seeking the feedback of the participants on key components of the programme to help us inform future work. We hypothesise that light to moderate structured exercise training is low risk and feasible in patients awaiting inpatient cardiac surgery. The study was approved by Health Research Authority and Heath and Care Research Wales (Yorkshire & the Humber- Bradford Leeds Research Ethics Committee: REC reference 23/YH/0255) on the 8th November 2023.

---

**Data availability statement:** No datasets were generated or analysed during the current study. All relevant data from this study will be made available upon study completion.

**Funding:** The author(s) received no specific funding for this work.

## Discussion

Multimodal prehabilitation could improve individual physical and mental wellbeing whilst awaiting inpatient cardiac surgery. Prehabilitation can provide individuals with a sense of ownership and control over their condition, improve their motivation and independence, and enhance their mental and physical recovery after surgery. Traditionally, patients waiting for cardiac surgery are discouraged from physical activity/ structured exercise training and receive limited information regarding their health. Appropriate physical and psychological support could improve their confidence to mobilise sooner after surgery. This may then facilitate earlier discharge leading to improved hospital bed utilisation and patient flow.

## Trial registration

ClinicalTrials.gov NCT06275100

## Introduction

Over 30,000 heart operations are performed every year in the UK; about a third are classified as urgent [1]. Conditions requiring urgent surgeries are not immediately life threatening but need to be dealt with in a timely fashion. In the cardiac surgery setting, a patient presenting to hospital with symptom such as chest pain is likely to be kept in hospital for further assessment and investigations; their condition is then discussed at multidisciplinary team meetings to determine further management.

During the Covid-19 pandemic, waiting times for coronary artery bypass graft (CABG) increased by 94% [1]. According to the National Institute for Cardiovascular Outcomes Research (NICOR), current waiting time between angiography and surgery for an urgent inpatient awaiting CABG is 11 days [1]. Median waiting times ranged from 7 to 24 days [1]. This is in contrast to the national guidelines stating that surgery should be carried out within 7 days [1]. Only 34% of acute patients receive their surgery within 7 days in 2019/2020 [1]. The delay could be due to a multitude of reasons such as waiting for anti-platelet therapy to diminish, optimisation of medical conditions, lack of beds in tertiary hospitals, or awaiting further investigations. Whilst in hospital, they receive limited information regarding their mental or physical health and are discouraged from leaving their bedside, spending 83% of their stay in a bed and 12% in a chair [2]. The reduced mobility and independence may eventually lead to Hospital-Associated Deconditioning (HAD); which impacts cognitive and physical function [3]. Deconditioning is associated with increased healthcare costs, extended hospital stays, decline in cognitive function and increased risk of mortality [3]. The impact of deconditioning on psychological health is understudied [3]. The longer waiting times, physical and psychological declines associated with HAD, has led clinicians and academics to consider the potential for optimising individual physical and psychological wellbeing prior to surgical intervention within the hospital environment.

The term prehabilitation has been used to describe interventions to improve physical and mental health prior to surgery. This could include pre-optimisation of medical conditions [4], exercise [5–9], education [5,7], respiratory muscle training [8–11], psychological [12,13] or nutritional [13,14] interventions. Regardless of the definition, prehabilitation is any intervention that prepares an individual, before an actual stress or physiological insult with the aim of improving their ability to recover. The Society of Enhanced Recovery after Cardiac Surgery (ERAS-C) has recommended that prehabilitation programme should include Nutrition (N), Exercise capacity (E) and Worry reduction (W)[13,15]. Evidence for the benefits of prehabilitation are well-established in patients undergoing orthopaedic or cancer surgery [16–18].

However, evidence within the cardiac surgical population is lacking. Prehabilitation trials are slow to develop in this group partly due to the concerns of adverse events amongst those with pre-existing and uncorrected heart conditions.

Acute in-patients are not traditionally offered prehabilitation for various reasons. Patients who are in hospital have traditionally been considered as acutely unwell and as such unable to participate in prehabilitation. However, each patient should be assessed individually throughout their admission as their ability to participate in various components of multimodal prehabilitation could change throughout their admission. In addition, it is thought that the short duration of the programme in this cohort may not have the desired impact. However, there is no evidence to support a minimum duration of prehabilitation for it to be deemed effective. We set out to design a multimodal programme based on the NEW interventions set out by ERAS-C [13,15].

## Prehabilitation- Nutrition (N)

Malnourished cardiac patients have a longer length of hospital stay, increased mortality and infection rates [19,20]. The prevalence of malnutrition amongst the cardiac surgery population is between 10–30% [19,21,22]. However, most of the malnutrition screening tools use BMI or recent weight loss to identify patients that are malnourished. This means that patients who have a raised BMI are potentially overlooked as being malnourished. In addition, weight loss is usually associated with a more chronic/ prolonged disease rather than an acute event. As such, malnutrition screening tools are usually not reflective of the acute cardiac surgical patients.

The average age of patients requiring cardiac surgery is 65 years old [1]. Nutritional support in the frail, elderly population has been shown to not only reduce 30-day mortality by over 50%, but also improve quality of life compared to standard hospital food [23]. Good preoperative nutritional practices have been shown to reduce post-operative hospital stay, mortality rates, complications and improve cardiac rehabilitation outcomes [14]. Prehabilitation trials should consider the feasibility including nutritional support to improve patient's outcome.

## Prehabilitation- Exercise (E) and Respiratory Muscular Training (RMT)

Different combinations of aerobic,[7–9] flexibility,[8] strength/ resistance training,[9] and inspiratory muscle training[9] have been trialed for prehabilitation. Despite the variations, it appears, patients still experience benefits from prehabilitation.

Exercise interventions have been shown to be safe and beneficial in elective patients awaiting CABG [5,7–9]. Steinmetz [8] and Sawatzky [7] reported no exercise related complications in elective CABG patients who underwent prehabilitation. Arthur [5] et al noted that more patients in the control group had worsening of cardiac status whilst waiting for surgery compared to those in the prehabilitation group. Hartog [9] et al noted similar rates of surgical escalation from elective to urgent/ emergency in prehabilitated (3.3%) and non prehabilitated (3.4%) patients. The same study found the prehabilitation group had lower incidence of atrial fibrillation [9]. However, all four studies had relatively small number of participants; the largest being Steinmetz [8] et al with 108 participants in the intervention group. To date, there are no published trials looking at the feasibility, safety or benefits of exercise on acute inpatients awaiting cardiac surgery.

Respiratory muscle training (RMT) has been trialed in prehabilitation to reduce the risk of post-operative pneumonia; one of the commonest complications after cardiac surgery. The incidence of pneumonia varies from 2–35% depending on the population and type of surgery; with patients post aortic dissection repair having the highest rate of chest infection [24–27]. Patients that suffer

from pneumonia are over 4 times more likely to die post-CABG, and experience longer hospital stays [24] which translates to increased cost of care. RMT can be delivered using Threshold Inspiratory Muscle Training (IMT) or Incentive Spirometer (IS). Threshold IMT and IS has been shown to improve post operative function [10,11,28] and reduce pulmonary complications [10,28]. Hulzebo et al noted that IS reduced post-operative pulmonary complications by almost 50%, pneumonia by 25% and in-patient stay by 1 day in high-risk CABG patients [10]. Interestingly, IMT and incentive spirometry (IS) not only reduced pulmonary complications but also increased the 6-minute walk test (6-MWT) [11,28], and IS decreased the reduction in 6-MWT performance in the first week post-surgery [28]. The improvements in pulmonary function tests, 6-MWT, quality of life and psychosocial parameters have been noted with IMT in just 5 days before surgery [11]. Incentive Spirometer (IS) is commonly used in hospital due to their low cost, ease of use and single patient use.

### Prehabilitation- Worry (W): Psychological factors

Cardiac surgery is associated with a negative impact on mental health with depression, anxiety and stress occurring in 20%, 23%, and 21% of patients respectively in waiting for CABG [29]. Preoperative anxiety is associated with an increased risk of mortality [29]. Although 33% experience an improvement in depression, anxiety and stress after CABG, new onset depression is diagnosed in 20% of patients in the months following surgery [30]. Preoperative depression is associated with decreased cardiac symptom relief, quicker return of symptoms, frequent hospitalisations, and increased mortality after the surgery whereas postoperative depression is associated with poor wound-healing and increased likelihood of infection [30].

Whilst the focus has always been skewed towards physical health, there is mounting evidence of inter-relationships between mental and physical health and wellbeing. Individuals with low resilience have more circulating norepinephrine and cortisol which is likely to explain the higher prevalence of hypertension in this population [31]. Strong emotions such as anger and negative emotions can precipitate MI [32]. The INTERHEART study found that psychosocial factors (depression, stress, critical life condition) increased the risk of MI by up to 3.9 times in females and 2.6 times in males, 2.9 times in the young, and 2.4 times in the elderly [33]. Mental health problems are often multifactorial and whilst difficult to pinpoint a single cause, the negative impact on overall health is significant. Despite this, psychological support is not routinely offered to the cardiac surgery population.

Ethical approval and trial registration:

### The PreP-ACe trial was ethically approved by the Health Research Authority and Health and Care

The PreP-ACe Trial has started recruitment in April 2024. This pilot feasibility trial will run for approximately 1 year. The PreP-ACe team will be the custodians of the final data set. Research Wales (Yorkshire & the Humber- Bradford Leeds Research Ethics Committee: REC reference 23/YH/0255) on the 8th November 2023. The trial sponsor is the University of Hull. The trial is registered at ClinicalTrials.gov (identifier: NCT06275100).

## Methodology

### Design

This is a mixed-methods single centre, prospective, single arm pilot feasibility trial.

### Setting

This trial will be conducted in Castle Hill Hospital (CHH), a tertiary hospital in United Kingdom. Assessments and interventions will be conducted on the cardiac surgery and

cardiology ward. Surgery and post-operative hospitalisation will take place in the same hospital.

## Study population

All patients admitted to CHH cardiology or cardiac surgery ward with heart disease will be considered and screened.

**General inclusion criteria.** All patients over the age of 18;
Admitted acutely with cardiac disease likely to require cardiac surgery;

**Able to speak, read, understand and provide informed consent in English;. General Exclusion criteria.** Impending surgery within 72 hours of arrival to hospital, e.g., aortic dissection, myxoma;

Surgery planned less than 5 days from recruitment;

Unable to participate in light to moderate exercise or unsupervised exercise due to functional issues, e.g., musculoskeletal conditions, stroke, Parkinson's disease or registered blind;

Cognitive impairment that would affect compliance;

Conditions that are unlikely to be treated with surgery, e.g., pericarditis, myocarditis, arrhythmias, heart blocks requiring pacemaker;

**Specific exclusion criteria.** Cardiac/ clinical instability such as:

- Recurrent chest pain since admission;

- Untreated decompensated heart failure;

- Resting tachycardia (HR > 100 bpm);

- Pre-syncopal or syncopal symptoms;

- Inability to provide written informed consent

## Randomisation and blinding

Those that meet the trial requirements will be given a patient information leaflet and invited to participate. Those that agree will be consented. As this is a single arm, pilot feasibility trial, it is not possible to mask group allocation from the participants or clinical staff. In addition, the primary objective is to assess the feasibility of prehabilitation in hospital.

## Data collection and management

The direct research team have participated in Information Governance training. Data will be used according to the General Data Protection Regulation (GDPR). Electronic information regarding the study will be stored on approved devices. Written hard copy data obtained from the study will be kept in a locked cabinet within a secure facility. All information collected will be anonymised. Individuals will not be identified through any reports or publications that result from the trial.

## Sample size

We did not perform a power calculation for sample size estimation as PreP-ACe is a pilot feasibility study. Some studies have suggested sample sizes of at least 10 participants [34,35]. Local audit data indicated that approximately 100 patients per year are admitted to CHH for acute inpatient surgery. Approximately 20% undergo surgery within the week. Based on these estimates, we aimed to recruit approximately 2 patients per month over a 12-month period. Therefore, our target sample size is 20 patients to account for 20% attrition.

## Patient and public involvement and engagement (PPIE)

During the study design phase, we interviewed 17 cardiac patients to gauge their interest in prehabilitation. 12 male and 5 female patients between 45–80 years old (average 65.7) were consulted. 13 patients were waiting for CABG, 2 valve surgery, 1 CABG and valve and 1 chronic dissection. All were keen to exercise whilst waiting for surgery except for one patient who felt that they would not be able to participate due to neuralgia. Patients felt that the interventions need to be tailored to each individual. There was mixed reaction to psycho-educational information and nutritional advice. 71% of patients felt that their anxiety could have been eased if they were provided more information. None of the patients had experience on meditation. We have used this information to guide the design of the psycho-education booklet. Patients felt they knew enough about nutrition and did not want information on nutrition content on the menu. As a result of the PPIE, we decided not to include nutritional information on food provided to the patients.

We consulted staff (physiotherapist, dietician, clinical psychologist, clinician and nurse) in CHH in the design of our intervention. The cardiology and cardiothoracic department currently do not have an allocated clinical psychologist. The hospital clinical psychologist is involved in the design of the psychoeducation booklet but will not review every participant individually. Our PPIE indicated that patients were happy to take supplements as part of their diet. However, after consultation with our local dietician, it was decided that due to the limited time between admission and surgery, supplements were unlikely to make a difference and the timescales for proposed dietary changes were felt to be unrealistic.

## Study procedures

The lead researcher (SR) is a clinician and MD student who will screen, assess and consent patients. After consenting, participants will be given information on psychological health (psychoeducation booklet) and given written information on accessing further psychological support if required. SR will conduct a series of baseline assessments (S1 Table) on the participants. SR will inform qualified physiotherapist (AH) regarding the recruitment. AH will assess their exercise abilities and prescribe exercise on an individualised basis. Participants who have dietary needs or concerns will be seen by a qualified hospital dietician. SR will be conducting the assessments and will seek feedback from the participants about the programme.

## Screening and assessment

Patients with conditions that are likely to require surgery will be screened for the above mentioned inclusion and exclusion criteria. Those that fit the study criteria will be approached as mentioned in the study procedures.

**Assessment.** We will perform several tests before the start of the intervention and during specified points during the intervention. Due to the nature of the admission, it is not possible to set an exact number of days as the end point for the intervention. As such the last day for the study will be the day before their surgery. We anticipate most patients to undergo surgery by day 7 and have set this as the day for reassessment. Patients that stay longer than 7 days before their surgery will be assessed for the third time.

1. Physical measurements include

- 6-minute walk test [36]

- Spirometry [37]

- Hand grip strength with dynamometer [38]

2. Assessing quality of life and mental health before the prehabilitation, after prehabilitation but before surgery

- EuroQoL (EQ-5D-5L) [39]

- Cardiac anxiety questionnaire (CAQ) [40,41]

3. Identify any safety concerns related to the intervention. This will be identified by patient reported diary and notes entry from the staff.

4. Information on post-operative complications and length of in-hospital stay will also be collected. We will collect complications such as chest infection, acute kidney injury, gastro intestinal, sternal wound infection, and post-operative arrhythmia.

5. All participants will be invited to provide a feedback on the programme with regards to the acceptability, practicality and enjoyment. We would also be seeking feedback on the psycho-education booklet.

The conduct of this study will not delay the course of patients' definitive treatment. In the event that the patient develops evidence of a worsening cardiac condition either related or unrelated to the intervention, they will be assessed by the medical team and treated appropriately either with alteration of medication or possibly have the surgery expedited. Once medically stable and a decision on their care is made by the medical team, the research team will assess the participant and consider if the study should continue for the participant. For example, if surgery is expedited, then the study would terminate at that point. However, if the decision is for further medical therapy or no change in the surgical plan, the study may continue with restrictions if the participant and medical team agrees. The decision for treatment options lie with the medical team directly involved in the patient's care. The researchers will not influence the decision regarding the patient's treatment.

## Study outcome measures

We will measure physical, psychological and quality of life outcomes at up to 3 time points depending on the duration of patient stay before the surgery. Outcome measures will be assessed after consent (baseline), Day 7 of intervention and Day 14 of intervention (S1 Fig). Patients that have surgery within 7 days of their intervention will have 2 assessments, those who have surgery after 7 days of assessment will have a total of 3 assessments. Throughout their in-hospital stay, we will also collect information on clinical outcomes.

The primary objective is to determine the feasibility of a prehabilitation programme in acute in patient population. We will be looking at the eligibility rates, enrolment/ recruitment rates, practicality of a prehabilitation programme in terms of logistical challenges to developing and implementing the programme (deliverability), uptake rates, tolerability, acceptability and completion rates of an inpatient prehabilitation programme. The decision for continuation, modification or early termination of the study will be based on the progression criteria set out in S2 Table. We will obtain feasibility data based on documentation from the researcher (eligibility, enrolment/recruitment, adherence, completion rates), physiotherapist documentation (frequency of intervention, completion rates, complications/challenges during exercise, practicality), patient feedback (practicality, tolerability, acceptability, enjoyment).

Secondary objectives are to identify any changes to their physical, psychological and quality of life outcomes. This is assessed via 6-MWT, hand grip strength [42], quality of life, spirometry and mental health before, during and after the intervention. We will be collecting information on adverse events, length of stay, complication(s) and readmission

during this study. This will give us signals for safety concerns and potential efficacy of the intervention.

Patients who decline to participate will be invited to provide a reason for their reluctance. Participants that withdraw from the study will be asked if there is a particular reason that could be addressed for the future.

## Interventions

We designed our interventions based on the NEW prehabilitation components recommended by Enhanced Recovery After Cardiac Surgery (ERAS-C) [15].

### Proposed intervention: Nutrition (N)

We consulted with dieticians locally and internationally; despite the prevalence of overnourished patients, active weight loss, 1–2 weeks before surgery is discouraged. We considered the use of nutritional supplement in our study and sought expert advice on the matter. Due to the short time frame between diagnosis and surgery, nutritional supplements were unlikely to be beneficial. We considered the possibility of hospital admission as an opportunity to educate patients on nutrition. However, during our PPIE, most patients felt that they had a healthy diet prior to admission and did not want education on nutrition. Due to the small number of participants, any education would be 1:1. The hospital dieticians are currently unable to support this need. As a result, we concluded that unfortunately, nutritional intervention would not be possible within the scope of our intervention.

### Proposed intervention: Exercise (E)

Participants will undergo a combination of aerobic, resistance/ strength training as well as balance and flexibility exercise. The exercise will be individualised according to the participant's ability. To ensure safety and convenience for participants, the interventions will be performed within the cardiology and cardiac surgery department. The exercise will be stopped if the patient develops symptoms such as angina (chest pain, or pain radiating down the left arm), arrhythmias or any syncopal symptoms. If the symptoms do not improve with rest, they will receive appropriate medical treatment. The exercise will then be stepped down in a gradual fashion with the minimum expectation of resistance exercise only. Participants who cannot perform at least resistance only exercise will be withdrawn from the study and the surgical team will be informed.

**Aerobic exercise.** Aerobic exercise will only be performed under 1:1 direct supervision by a trained exercise physiologist aiming for a low to moderate exercise intensity. This will range from 40–70% heart rate reserve (%HRR) equivalent to a Borg Rating of Perceived Exertion (RPE) 10–14 based on the British Association for Cardiovascular Prevention and Rehabilitation (BACPR) recommendations. The exercise physiologist will aim to progress the exercise duration and intensity (within the low to moderate range) in subsequent training sessions. At any stage of the intervention, if the patient develops symptoms as previously described, the intervention will be stopped, they will be re-assessed and treated by the medical team.

Aerobic exercise will be performed on a cycle ergometer (Watt-bike) and each patient will be connected to an electrocardiograph (ECG) throughout the exercise. The exercise training protocol will employ an interval approach, alternating every minute between low to moderate intensity aerobic exercise. The initial intensity will be based on initial assessment. The duration of the training sessions will increase (to a maximum of 20 minutes; 10 x 1 minute light intensity and 10 x 1 minute moderate intensity), and intensity will increase to a maximum

of 70% HRR or RPE 14 (moderate intensity) according to individual ability and the window between hospitalisation and surgery. Aerobic exercise will be conducted on 2–3 occasions per week dependent upon individual ability, with a recovery day between training sessions. Patients unable to use the Watt-bikes will be provided with alternative exercise such as seated pedals or walking. The number of training sessions will be recorded alongside workload, heart rate and RPE during each stage.

**Resistance/ strength exercise.** Patients will be given the option of an in-hospital phone-based digital application (Wibbi https://wibbi.com/) or printed information on how to perform resistance exercise by their bedside. The application has not been used previously for patients awaiting cardiac surgery so the physiotherapy team will tailor exercises for the patients and set the targets. The participants will have an individualised log in for the application. They will click onto the exercise prescribed and once they have completed the task they will record it as a goal achieved. The physiotherapist are able to monitor the frequency of participation. Patients who are unable to perform the aerobic exercise (e.g., due to physical limitation or symptoms) will be asked to perform resistance exercise only. The frequency of exercise will be individualised according to patient's ability and fitness level.

**Respiratory Muscular Training (RMT).** As part of the intervention, patients will be given an IS before surgery with appropriate educational support. IS is selected over Threshold IMT as IS is easier to use and available in the hospital. The importance of RMT will be explained. Patient can access RMT using a smart-phone-based app. The target will be for patients to do 3 sets (consisting of 6 breath each), 5 times a day. Preoperative training should also aid better post-operative compliance. Engagement with the RMT component will be monitored.

## Proposed intervention: Worry (W)

All study participants will be given a booklet containing psychoeducation information. The booklet aims to explain the emotions they may feel whilst being in hospital, symptoms of anxiety, suggest how to manage anxiety, behavioural strategies that may help, and signposting if they require further support. Through reading this booklet, participants are encouraged to engage in psychoeducation exercises and develop emotional wellbeing strategies. In addition to this, the participants will be given access to audio on guided meditation and relaxation. Both the booklet and audio are designed with input from the in-house clinical psychologist.

Patients will be asked to score themselves on CAQ and EQ-5D-5L on day 0, day 14 or day before surgery. During the second assessment of CAQ and EQ-5D-5L, patients will be asked if they have read or used either the psycho-education booklet or audio. Hospital staff will be alerted of participants that score highly and considered high risk. In addition, they will be referred to the in-hospital psychiatry liaison team.

## End of study

The last day of the intervention will be the day before the surgery. At the end of the study, a semi-structured interview will be conducted to obtain feedback on the programme. The research team will observe the patient for complications such as infections, arrhythmias, acute kidney injury, mortality as well as length of hospital stay. Participants will be interviewed at the end of the intervention to explore their opinions on the acceptability, practicality, barriers and enablers as well as a feedback on the intervention.

## Statistical analysis

We will report feasibility outcomes (participant eligibility, acceptability, recruitment rates, completion rates) in percentages and/ or counts. We will be describing barriers to implementing a

prehabilitation programme in acute inpatients waiting for cardiac surgery. Distribution of data will be described using median and interquartile range. Changes in the physical, psychological quality of life measures for each patient will be plotted on graphs. Data will be analysed using StataMP 18. Qualitative data from semi-structured interviews will be transcribed verbatim. Complication rates and adverse events will be reported as percentages and/ or count. Pilot feasibility data may be used to inform a fully powered randomised controlled trial.

## Data monitoring, adverse events and auditing

A trial management steering group will meet every four months to discuss trial progression, data monitoring, trial conduct and safety considerations. The trial management steering group will consist of key collaborators. Adverse events that may be attributed to any of the intervention will be monitored by the PreP-ACe researcher (SR), and physiotherapist (AH). The events are then reported to the project supervisor (LI) and discussed with an independent clinician not involved in the study.

We anticipate some patients with cardiac conditions may experience symptoms of postural hypotension which may impact their ability to perform the test. If this occurs, it will be documented by the assessor and patients are told to rest before trying the test again. If they are unable to perform the test at all, they will be withdrawn from the study. We will record adverse and serious adverse events. In the event that the patient develop evidence of worsening cardiac condition either related or unrelated to the intervention, they will be assessed by the medical team and treated appropriately either with alteration of medication or possibly have the surgery expedited. The decision for treatment options lie with the medical team directly involved in the patient's care. The researchers will not influence the decision regarding the patient's treatment.

## Dissemination and impact

Throughout the trial, the progress will be monitored by project supervisors (LI, GD, DY). At the end of the study, the results will be published in peer-reviewed journals and presented at scientific meetings.

PreP-ACe trial could have an impact on

1. Development of future studies on prehabilitation including informing randomised controlled trials

2. Clinician perceptions on prehabilitation

3. Patient's perception of prehabilitation and understanding of their own health. Prehabilitation could give back a sense of control and ownership of the disease to the patient.

4. Development of future services to improve patient care such as provision of physiotherapist, clinical psychologist and dietician

5. Challenging the current 'standard care' for patients who are require urgent cardiac surgery

The NHS is constantly under pressure to perform better, and do more with less resources. The Covid-19 pandemic has not helped the situation but made it even more challenging. One of the limitations for service expansion is the availability of beds. The questions that this study will hopefully answer includes:

1. Is in-hospital prehabilitation feasible in cardiac patients awaiting acute surgery?

2. Does in-hospital prehabilitation lead to adverse outcome(s) in this population?

If the prehabilitation is safe in this population, the next question that needs to be addressed is; do the patients need to be in hospital waiting for their cardiac surgery? What is the purpose of their inpatient stay? Perhaps, an idea for a 'remote acute list' where patients are sent home with information about their condition, plans and a date for investigation(s) and surgery. This will in turn free up bed spaces to allow more surgeries to be done.

## Supporting information

**S1 Figure. SPIRIT schedule for enrolment, intervention and assessment to the PreP-ACe pilot feasibility study.**
(TIFF)

**S1 Table. Outcome measures and time point assessments for individuals recruited to PreP-ACe.**
(PDF)

**S2 Table. Stop-go Criteria.**
(PDF)

**S1 File. SPIRIT checklist.**
(PDF)

**S2 File. TIDieR checklist.**
(PDF)

**S3 File. Informed consent.**
(PDF)

## Acknowledgements

We would like to thank all of the co-investigators involved in this trial who have worked collaboratively to develop this protocol. We are all invested in improving clinical outcomes for citizens awaiting surgical intervention in the region.

## Author contributions

**Conceptualization:** Sarah Raut, Dumbor Ngaage, Lee Ingle.

**Methodology:** Sarah Raut, Aaron Hales, Maureen Twiddy, Lili Dixon, Dumbor Ngaage, Lee Ingle.

**Supervision:** David Yates, Gerard Danjoux, Lee Ingle.

**Writing – original draft:** Sarah Raut.

**Writing – review & editing:** Sarah Raut, Aaron Hales, Maureen Twiddy, Lili Dixon, Dumbor Ngaage, David Yates, Gerard Danjoux, Lee Ingle.

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
