## [Decision Letter · Decision Letter 0]

6 Sep 2024

PONE-D-24-27101Multimodal Prehabilitation in People Awaiting Acute Inpatient Cardiac Surgery:  Study Protocol for a Pilot Feasibility Trial (PreP-ACe)PLOS ONE

Dear Dr. Raut,

Thank you for submitting your manuscript to PLOS ONE. After careful consideration, we feel that it has merit but does not fully meet PLOS ONE’s publication criteria as it currently stands. Therefore, we invite you to submit a revised version of the manuscript that addresses the points raised during the review process.

We look forward to receiving your revised manuscript.

Kind regards,

Yashendra Sethi

Academic Editor

PLOS ONE

Journal Requirements:

3. We note that the original protocol that you have uploaded as a Supporting Information file contains an institutional logo. As this logo is likely copyrighted, we ask that you please remove it from this file and upload an updated version upon resubmission.

Reviewers' comments:

Reviewer's Responses to Questions

**Comments to the Author**

1. Does the manuscript provide a valid rationale for the proposed study, with clearly identified and justified research questions?

Reviewer #1: Yes

Reviewer #2: Yes

2. Is the protocol technically sound and planned in a manner that will lead to a meaningful outcome and allow testing the stated hypotheses?

Reviewer #1: Yes

Reviewer #2: Partly

3. Is the methodology feasible and described in sufficient detail to allow the work to be replicable?

Reviewer #1: Yes

Reviewer #2: Yes

4. Have the authors described where all data underlying the findings will be made available when the study is complete?

Reviewer #1: Yes

Reviewer #2: Yes

5. Is the manuscript presented in an intelligible fashion and written in standard English?

Reviewer #1: Yes

Reviewer #2: Yes

6. Review Comments to the Author

You may also provide optional suggestions and comments to authors that they might find helpful in planning their study.

Reviewer #1: Thank you for the invitation to review this interesting study protocol. The rationale is clearly articulated that this is a novel intervention and important area of research. I have made a few comments below for your consideration.

Sample size - I understand approximately 100 patients were admitted to CHH per year and would be available for recruitment but what proportion of these would be likely to be eligible e.g. how many had a condition unlikely to be treated with surgery? How many underwent surgery in less than 5 days post-admission?

PPIE - As written it in unclear how feedback from PPIE members influenced the study or intervention design - for example did PPIE members feed into the content of the exercise programme? It may also be worth commenting on how representative PPIE members were of the population e.g. considering sex, race, economic status. Will PPIE members have a role throughout the trial?

Data collection - Would patients usually be referred to cardiac rehab on discharge from inpatient care? - if so it would be interesting to collect data on uptake. It is possible that those who receive prehab refuse cardiac rehab (this was observed by Greening et al 2018 with regards to pulmonary rehab), this would be an unintended consequence.

Data collection - the content of the intervention and the outcome measures do not appear to be completely aligned - The exercise intervention includes balance and strength training yet there is no outcome measures to assess strength (although grip strength could be a proxy for lower-limb strength) or balance (e.g. TUG, SLS). Did the authors considered assessing MIP as IMT is part of the intervention?

Intervention - please consider using thee TiDiER checklist. Currently there is no description of the balance/flexibility exercises to be performed.

In a feasibility trial I would expect to see "stop go criteria" or progression criteria?

End of study and Statistical analysis - this is the first time a qualitative component is mentioned? In the study design it should be clear that a mixed-methods approach is being applied. Who is conducting the interviews? Where? What is the purpose of the interviews? What analysis will be applied?

Reviewer #2: As a pilot feasibility study, no formal sample size calculation is required. However, since the primary objective of this study is to determine the feasibility of a prehabilitation programme, it is expected to see how the feasibility will be quantified. For example, how big of the enrollment rate or completion rate would establish the feasibility. Even though the statistical analysis for such a pilot study is mainly descriptive, it would be better to have more details such as all 95% confidence intervals will be reported for any rates. There would be longitudinal data (such as QoL) collected over time and hence proper models for such data should be used in order to better estimate the change in QoL after the programme. Consultation with and/or the involvement of a professional biostatistician in the study team is strongly recommended.

7. PLOS authors have the option to publish the peer review history of their article (what does this mean? ). If published, this will include your full peer review and any attached files.

**Do you want your identity to be public for this peer review?** For information about this choice, including consent withdrawal, please see our Privacy Policy .

Reviewer #1: **Yes: ** Samantha L Harrison

Reviewer #2: No

---

## [Author Response · Author response to Decision Letter 1]

12 Oct 2024

Reviewer #1: Thank you for the invitation to review this interesting study protocol. The rationale is clearly articulated that this is a novel intervention and important area of research. I have made a few comments below for your consideration.Sample size - I understand approximately 100 patients were admitted to CHH per year and would be available for recruitment but what proportion of these would be likely to be eligible e.g. how many had a condition unlikely to be treated with surgery? How many underwent surgery in less than 5 days post-admission?

We are currently looking to publish data from our scoping work as such I am unable to provide too many detail. Approximately 100 patients are admitted directly to CHH for acute cardiac surgery in a year. 22% underwent surgery within the week (as per NICOR), of the remaining patients, some would have been too unwell to participate in the program or require medical optimisation that would impinge on their ability to participate in the program. PPIE - As written it in unclear how feedback from PPIE members influenced the study or intervention design - for example did PPIE members feed into the content of the exercise programme? It may also be worth commenting on how representative PPIE members were of the population e.g. considering sex, race, economic status. Will PPIE members have a role throughout the trial?

This is now included in the PPIE section. With a breakdown of the patient ages and type of surgery. We did not collect data on economic status. Due to PPIE and staff feedback, we have removed the nutritional element of the study. This has been explained in the intervention section (nutrition). There may be an opportunity for education classes in the future but this is not feasible with our current trial. Our psychoeducation booklet is designed to provide information on their admission as well as meditation.

Data collection - Would patients usually be referred to cardiac rehab on discharge from inpatient care? - if so it would be interesting to collect data on uptake. It is possible that those who receive prehab refuse cardiac rehab (this was observed by Greening et al 2018 with regards to pulmonary rehab), this would be an unintended consequence.

Currently, all patients would be referred to cardiac rehab post discharge. This would be an interesting point and we can certainly try to follow these patients up post discharge.

Data collection - the content of the intervention and the outcome measures do not appear to be completely aligned - The exercise intervention includes balance and strength training yet there is no outcome measures to assess strength (although grip strength could be a proxy for lower-limb strength) or balance (e.g. TUG, SLS). Did the authors considered assessing MIP as IMT is part of the intervention?

When designing the study, we did initially consider TUG. However, after much consideration, this was removed due to concerns of postural hypotension in this population. Again, with SLS, we were not sure about the safety of performing this test in acute inpatients. It maybe that this will be added in the future study as we gain more insight and confidence.

Yes, we did consider MIP but the equipment was not readily available. Due to the restricted budget and available personel, we had to rationalise our interventions and outcome measures. If we gain future funding, this could change.

Intervention - please consider using thee TiDiER checklist. Currently there is no description of the balance/flexibility exercises to be performed.

TiDiER checklist included. The balance and flexibility exercise will be based on the physiotherapist assessment.

In a feasibility trial I would expect to see "stop go criteria" or progression criteria?

Agreed. We have added criteria as you indicate.

End of study and Statistical analysis - this is the first time a qualitative component is mentioned? In the study design it should be clear that a mixed-methods approach is being applied. Who is conducting the interviews? Where? What is the purpose of the interviews? What analysis will be applied?

The method section of the abstract does mention seeking feedback from the participants. On figure 1: SPIRIT schedule, there is a section on Post study feedback at the end of the intervention. The authors accept this could be made clearer; as such, a section is added under study procedures and study outcomes. The researcher SR will be seeking feedback from the participants at then end of the intervention to ascertain practicality, acceptability and enjoyment of the program as well as feedback on the psychoeducation booklet. Whenever possible, data will be provided as percentages or counts. However, free-prose responses will be transcribed verbatim.

Reviewer #2: As a pilot feasibility study, no formal sample size calculation is required. However, since the primary objective of this study is to determine the feasibility of a prehabilitation programme, it is expected to see how the feasibility will be quantified. For example, how big of the enrollment rate or completion rate would establish the feasibility. Even though the statistical analysis for such a pilot study is mainly descriptive, it would be better to have more details such as all 95% confidence intervals will be reported for any rates. There would be longitudinal data (such as QoL) collected over time and hence proper models for such data should be used in order to better estimate the change in QoL after the programme. Consultation with and/or the involvement of a professional biostatistician in the study team is strongly recommended.

Thank you. We will look to include a biostatistician as the project progresses and we determine whether we wish to pursue a full RCT. In relation to your example, we are currently not collecting long-term QoL data. The end of study will be the day before the surgery. It would be interesting for future study to perform a case- control study to assess QoL change after the program.

---

## [Decision Letter · Decision Letter 1]

18 Nov 2024

PONE-D-24-27101R1Multimodal Prehabilitation in People Awaiting Acute Inpatient Cardiac Surgery:  Study Protocol for a Pilot Feasibility Trial (PreP-ACe)PLOS ONE

Dear Dr. Raut,

Thank you for submitting your manuscript to PLOS ONE. After careful consideration, we feel that it has merit but does not fully meet PLOS ONE’s publication criteria as it currently stands. Therefore, we invite you to submit a revised version of the manuscript that addresses the points raised during the review process.

**ACADEMIC EDITOR: The reviewer have again raised some minor concerns. Please have a look and address them soon.**

We look forward to receiving your revised manuscript.

Kind regards,

Yashendra Sethi

Academic Editor

PLOS ONE

Journal Requirements:

Reviewers' comments:

Reviewer's Responses to Questions

**Comments to the Author**

1. Does the manuscript provide a valid rationale for the proposed study, with clearly identified and justified research questions?

Reviewer #3: Yes

2. Is the protocol technically sound and planned in a manner that will lead to a meaningful outcome and allow testing the stated hypotheses?

Reviewer #3: Yes

3. Is the methodology feasible and described in sufficient detail to allow the work to be replicable?

Reviewer #3: Yes

4. Have the authors described where all data underlying the findings will be made available when the study is complete?

Reviewer #3: Yes

5. Is the manuscript presented in an intelligible fashion and written in standard English?

Reviewer #3: Yes

6. Review Comments to the Author

You may also provide optional suggestions and comments to authors that they might find helpful in planning their study.

Reviewer #3: Well thought out and good idea to consider prehabilitation in selected individuals waiting for cardiac surgery. I greatly appreciate the authors addressing previous reviewer comments. I would recommend to include the demographics of the patient population - age, sex, BMI; some of the common medical comorbidities that are associated with cardiac surgery - HTN, DM2, PAD, AKI or CKD, HLD, COPD, or on O2 at baseline; albumin level; LV function prior to surgery, presence of any IV or subcutaneous antiplatelets agents or anticoagulants, risk of falls/fragility, are they functionally independent prior to this admission. If they don't have these medical comorbidities, these patients would tend to participate more in the prehabilitation program and you would only have stable patients who would do well in the long run regardless if they undergo prehabilitation or not. It would be good to see these data in the feasibility study.

It is uncommon to wait beyond 7 days for an urgent cardiac surgery in US or some Asian countries.

"Only 34% of acute patients receive their surgery within 7 days in 2019/2020." Things are different post COVID - you can consider what's your current % of these patients getting their surgery within 7 days. If this % way higher than before, you may have difficulty recruiting patients beyond the feasibility trial.

7. PLOS authors have the option to publish the peer review history of their article (what does this mean? ). If published, this will include your full peer review and any attached files.

**Do you want your identity to be public for this peer review?** For information about this choice, including consent withdrawal, please see our Privacy Policy .

Reviewer #3: No

---

## [Author Response · Author response to Decision Letter 2]

20 Nov 2024

Dear Reviewer,

Thank you very much for your time and feedback on my protocol- Multimodal prehabilitation in People awaiting acute inpatient cardiac surgery: study protocol for a pilot feasibility trial (PreP-ACe). I have taken your comments into consideration and amended where appropriate. Please see below the responses to your feedback.

Reviewer #3: Well thought out and good idea to consider prehabilitation in selected individuals waiting for cardiac surgery. I greatly appreciate the authors addressing previous reviewer comments. I would recommend to include the demographics of the patient population - age, sex, BMI; some of the common medical comorbidities that are associated with cardiac surgery - HTN, DM2, PAD, AKI or CKD, HLD, COPD, or on O2 at baseline; albumin level; LV function prior to surgery, presence of any IV or subcutaneous antiplatelets agents or anticoagulants, risk of falls/fragility, are they functionally independent prior to this admission. If they don't have these medical comorbidities, these patients would tend to participate more in the prehabilitation program and you would only have stable patients who would do well in the long run regardless if they undergo prehabilitation or not. It would be good to see these data in the feasibility study.

Thank you very much for your feedback. We will consider all acute patients waiting for inpatient cardiac surgery. When we report the outcome of the feasibility study, we will include age, sex, BMI, co-morbidities etc as stated above. We do expect patients who participate to have a degree of stability (eg: asymptomatic at rest, sepsis with ongoing treatment).

It is uncommon to wait beyond 7 days for an urgent cardiac surgery in US or some Asian countries.

This study is conducted in the UK where the main healthcare provider is publicly funded. Unfortunately, waiting times are a feature of public funded healthcare."Only 34% of acute patients receive their surgery within 7 days in 2019/2020." Things are different post COVID - you can consider what's your current % of these patients getting their surgery within 7 days. If this % way higher than before, you may have difficulty recruiting patients beyond the feasibility trial.

Thank you very much for your feedback. We did consider this. As such we have done some scoping work. We analysed patients who came to our hospital in 2022 and tried to describe them based on their presentation, co-morbidities, type of surgery and outcome. As this is a separate piece of work currently under review, we are unable to make reference to this in the protocol.

---

## [Editor Report · Decision Letter 2]

26 Nov 2024

Multimodal Prehabilitation in People Awaiting Acute Inpatient Cardiac Surgery:  Study Protocol for a Pilot Feasibility Trial (PreP-ACe)

PONE-D-24-27101R2

Dear Dr. Raut,

We’re pleased to inform you that your manuscript has been judged scientifically suitable for publication and will be formally accepted for publication once it meets all outstanding technical requirements.

Kind regards,

Yashendra Sethi

Academic Editor

PLOS ONE
---

## [Editor Report · Acceptance letter]

PONE-D-24-27101R2

PLOS ONE

Dear Dr. Raut,

I'm pleased to inform you that your manuscript has been deemed suitable for publication in PLOS ONE. Congratulations! Your manuscript is now being handed over to our production team.

Kind regards,

on behalf of

Dr. Yashendra Sethi

Academic Editor

PLOS ONE